# Wafer-scale high-κ dielectrics for two-dimensional circuits via van der Waals integration

Zheyi Lu[1], Yang Chen[1], Weiqi Dang[2], Lingan Kong[1], Quanyang Tao[1], Likuan Ma[1], Donglin Lu[1], Liting Liu[1], Wanying Li[1], Zhiwei Li[1], Xiao Liu[1], Yiliu Wang[1], Xidong Duan ®[2], Lei Liao ®[1] & Yuan Liu ®[1]✉

The practical application of two-dimensional (2D) semiconductors for high-performance electronics requires the integration with large-scale and high-quality dielectrics—which however have been challenging to deposit to date, owing to their dangling-bonds-free surface. Here, we report a dry dielectric integration strategy that enables the transfer of wafer-scale and high-κ dielectrics on top of 2D semiconductors. By utilizing an ultra-thin buffer layer, sub-3 nm thin $Al_2O_3$ or $HfO_2$ dielectrics could be pre-deposited and then mechanically dry-transferred on top of $MoS_2$ monolayers. The transferred ultra-thin dielectric film could retain wafer-scale flatness and uniformity without any cracks, demonstrating a capacitance up to 2.8 μF/cm², equivalent oxide thickness down to 1.2 nm, and leakage currents of ~$10^{-7}$ A/cm². The fabricated top-gate $MoS_2$ transistors showed intrinsic properties without doping effects, exhibiting on-off ratios of ~$10^7$, subthreshold swing down to 68 mV/dec, and lowest interface states of $7.6 \times 10^9$ cm$^{-2}$ eV$^{-1}$. We also show that the scalable top-gate arrays can be used to construct functional logic gates. Our study provides a feasible route towards the vdW integration of high-κ dielectric films using an industry-compatible ALD process with well-controlled thickness, uniformity and scalability.

Two-dimensional (2D) semiconductors are considered as promising candidates for next-generation electronic devices because of their atomically thin body thickness with superior gate controllability, dangling-bonds-free surface as well as higher carrier mobility[1–5]. To construct large-scale circuits, the integration of high-quality dielectric on 2D semiconductor surface is of great importance. In modern silicon microelectronics, atomic layer deposition (ALD) process has been widely applied for depositing high-κ dielectrics (e.g., $Al_2O_3$, $HfO_2$) on semiconducting channel, owing to its ability to scalable integrate high-quality films with well-controlled thickness[6,7]. However, applying the existing state-of-the-art ALD process on 2D semiconductors is

challenging due to their pristine surface without any dangling bonds for chemical reaction[8–10].

Considerable efforts have been devoted to deposit high-quality dielectric on 2D surface through interface engineering. Early attempts modified the 2D surface through plasma pre-treatment by intentionally creating defect sites or new dangling bonds (e.g., S vacancy or Mo-O bonds for $MoS_2$), which can be activated to enable following ALD process. However, the modification of 2D surface typically introduce trap states and damages to delicate 2D lattice, degrading their intrinsic properties[11–13]. Alternatively, thin polymer or molecular layer could serve as a buffer to support the nucleation of high-κ dielectrics on 2D

[1]Key Laboratory for Micro-Nano Optoelectronic Devices of Ministry of Education, School of Physics and Electronics, Hunan University, Changsha 410082, China. [2]Hunan Key Laboratory of Two-Dimensional Materials, State Key Laboratory for Chemo/Biosensing and Chemometrics, College of Chemistry and Chemical Engineering, Hunan University, Changsha 410082, China. ✉e-mail: yuanliuhnu@hnu.edu.cn

surface without altering their intrinsic structures and properties[14,15]. However, these buffer layers typically exhibit poor stability and low dielectric constant, serving as a series capacitance and weakening the overall gate controllability. Recently, van der Waals insulators (such as BN, mica, $CaF_2$, perovskite)[16–22] have been explored as gate dielectric for 2D transistors, because they could form van der Waals (vdW) interfaces with minimized trap states and damages to the underlying 2D semiconductors. However, synthesizing large-scale vdW dielectric is challenging, especially for multilayer dielectrics with desired thickness[16,18]. In addition, recent study suggested that these layered dielectrics typically demonstrate poor insulating behavior and large leakage current owning to vdW gap within vertical direction, limiting their practical application for low-power electronics[23]. Hence, it remains a critical challenge to integrate large-scale and high-quality dielectric on 2D semiconductors, posing an important technological challenge for pushing the performance limit of 2D transistor as well as the practical application of 2D-based integrated circuits.

Here, we report a dry dielectric integration strategy that could transfer wafer-scale high-κ dielectric on top of 2D semiconductors, therefore could maintain their delicate lattice and intrinsic properties. By utilizing an ultra-thin PVA (polyvinyl alcohol) as sacrifice layer, sub-3 nm thick $Al_2O_3$ or $HfO_2$ dielectrics could be pre-deposited and then mechanically dry-released and dry-laminated on top of wafer-scale $MoS_2$ monolayers. Owning to the low strain induced during dry-lamination process (compared to conventional wet-transfer process), the transferred ultra-thin dielectric film could retain wafer-scale flatness and uniformity without any cracks, demonstrating high capacitance of $2.8 \, \mu F/cm^2$, small equivalent oxide thickness (EOT) of 1.2 nm, low leakage current of $10^{-7} \, A/cm^2$, and high breakdown field of 6 MV/cm, consistent with the as-deposited high-κ dielectrics. Taking the advantage of weakly coupled vdW dielectric interface, the fabricated top-gate $MoS_2$ transistor arrays exhibit intrinsic properties without any doping effect, demonstrating high on-off ratio of $10^7$, low subthreshold swing of 68 mV/dec, and lowest interface states of $7.6 \times 10^9 \, cm^{-2} \, eV^{-1}$. This is in great contrast to conventional previous ALD process with strong doping effect to 2D channel with much reduced on-off ratio[24]. The scalable fabrication of large-area top-gate transistors further enabled the construction of functional logic gates and computational circuits, including an NOT, NAND, NOR, AND and XOR gates. Our study not only demonstrate a vdW approach to integrate high-κ dielectrics on 2D semiconductors using industry-compatible ALD process with well-controlled thickness and uniformity, but also provide a dry and wafer-scale integration process of different dielectrics and bulk materials. It may also provide exciting implications for various delicate materials beyond 2D semiconductors (such as perovskite, organic monolayers) that are previously plagued by the ill-defined dielectric-semiconductor interface (e.g., with the dangling-bonds-free surface) or are highly prone to degradation during the dielectric deposition process, therefore enable the investigation of fundamental physics and high-performance devices not previous possible.

## Results
### Dry transfer process of wafer-scale high-κ dielectrics
The transfer processes of high-κ dielectric are schematically illustrated in Fig. 1a–c, and also detailed in Methods section. In brief, ultra-thin PVA layer (9 nm thick) is first spin-coated on top of a functionalized silicon wafer, serving as a buffer layer for the following ALD process. Next, 2-inch-size $Al_2O_3$ with various thickness are directly deposited on the PVA surface through ALD process, and the deposited thickness can be well-controlled through deposition cycles. The PVA/$Al_2O_3$ stack could then be mechanically peeled-off from the silicon wafer using a thermal release tape, and the exposed bottom PVA layer could be dry-etched through $O_2$ plasma, as schematics illustrated in Fig. 1b. The dry-peeling and dry-etching processes here is important to avoid solution-induced random strains during wet-etching process, and is

essential to ensure a continuous dielectric film in wafer-scale without any cracks and residues[25,26]. Finally, the released $Al_2O_3$ film could be transferred and laminated onto large-scale $MoS_2$ monolayer (grown by chemical vapor deposition) by heating the sample to decrease the adhesion force between tape and high-κ dielectric film (Fig. 1c). The corresponding optical images of wafer-scale transfer process is also shown in Fig. 1a–c and Supplementary Movie 1.

The use of thin PVA buffer layer here is essential to transfer wafer-scale and uniform high-κ dielectric, due to the following reasons. First, PVA has a relatively high melting temperature of 230 °C (ref. 27), which could still demonstrate low adhesion force after the ALD process and can be mechanically peeled-off. For other conventional polymer buffers (such as polymethyl methacrylate (PMMA), polypropylene carbonate (PPC)), they will be either partially melted or deformed after the ALD process and can not be released from substrate, as shown in Supplementary Fig. 1. Second, the spin-coated PVA buffer is thin enough (~9 nm), and could be easily dry-etched using gentle plasma treatment. More importantly, the thin PVA layer still demonstrates atomic flat surface (0.36 nm roughness, Supplementary Fig. 2) while retaining sufficient dangling bonds for following ALD process, which is essential to deposit ultra-thin and flat dielectric film. For other 2D buffer layers such as graphene, although they could sustain higher deposition temperature for emerging dielectric, sub-5 nm thick and high-quality dielectric is challenging to deposited owning to its dangling-bonds-free surface[10]. Third, strain is unavoidable during transfer process during the bending of the holding substrate, especially for wafer-scale transfer of dielectric film with brittle lattice. Using 9 nm thin PVA layer here, the applied strain during the dry-transfer process could also be minimized since strain is proportional to the substrate thickness, leading to uniform and wafer-scale dielectric layer without any cracks, as shown in Supplementary Fig. 3. We also applied the XPS (X-ray photoelectron spectroscopy) to examine the properties of $Al_2O_3$ during our transfer process, where consistent XPS peaks are observed, as shown in Supplementary Fig. 4.

Figure 1d shows the AFM measurement of the $Al_2O_3$ bottom surface after dry peeling-off, where crack-free and flat surface is observed with small root-mean-square (RMS) roughness of 0.3 nm. The transferred film is not only uniform in small regions, but also demonstrates flat surface across the whole transferred film in wafer size. To demonstrate this, we have contacted AFM measurement over hundreds of locations across a 2-inch transferred film, and each location have a scanning area of $10 \, \mu m \times 10 \, \mu m$ (with a spacing of ~2 mm between two adjacent locations). The RMS of each location could be obtained by analyzing the corresponding AFM height profile, where the 100 different RMS roughness data are plotted in two-dimensional graph in Fig. 1e. Furthermore, stair meter is applied to measure the surface morphology of $Al_2O_3$, demonstrating flat surface across millimeter region, as shown in Supplementary Fig. 5.

Furthermore, the demonstrated thin film transfer process is not only limit to $Al_2O_3$, and could be well extended to other oxide thin films including insulating $HfO_2$, semiconducting IGZO (indium gallium zinc oxide), and conducting ITO (indium tin oxide), as long as they can by deposited on the sacrifice PVA buffer, as shown in Supplementary Fig. 6. Here, as a proof-of-concept illustration, 2-inch-size $HfO_2$ and $Al_2O_3$ films are layer-by-layer transferred together, demonstrate a vdW $Al_2O_3/HfO_2/Al_2O_3$ superlattice in wafer scale, as shown in Fig. 1f. Using this approach, building up more complex heterostructure of oxide materials with various functions could be an interest topic for future investigation without previous limits of lattice matching or process compatibility.

### Dielectric properties of transferred oxide film
In order to study the dielectric properties of the transferred dielectric films, we have fabricated metal-insulator-metal structure (using transferred dielectric) and conducted corresponding $I–V$ measurement and $C–V$ measurement. As illustrated in Fig. 2a, the MIM device is fabricated

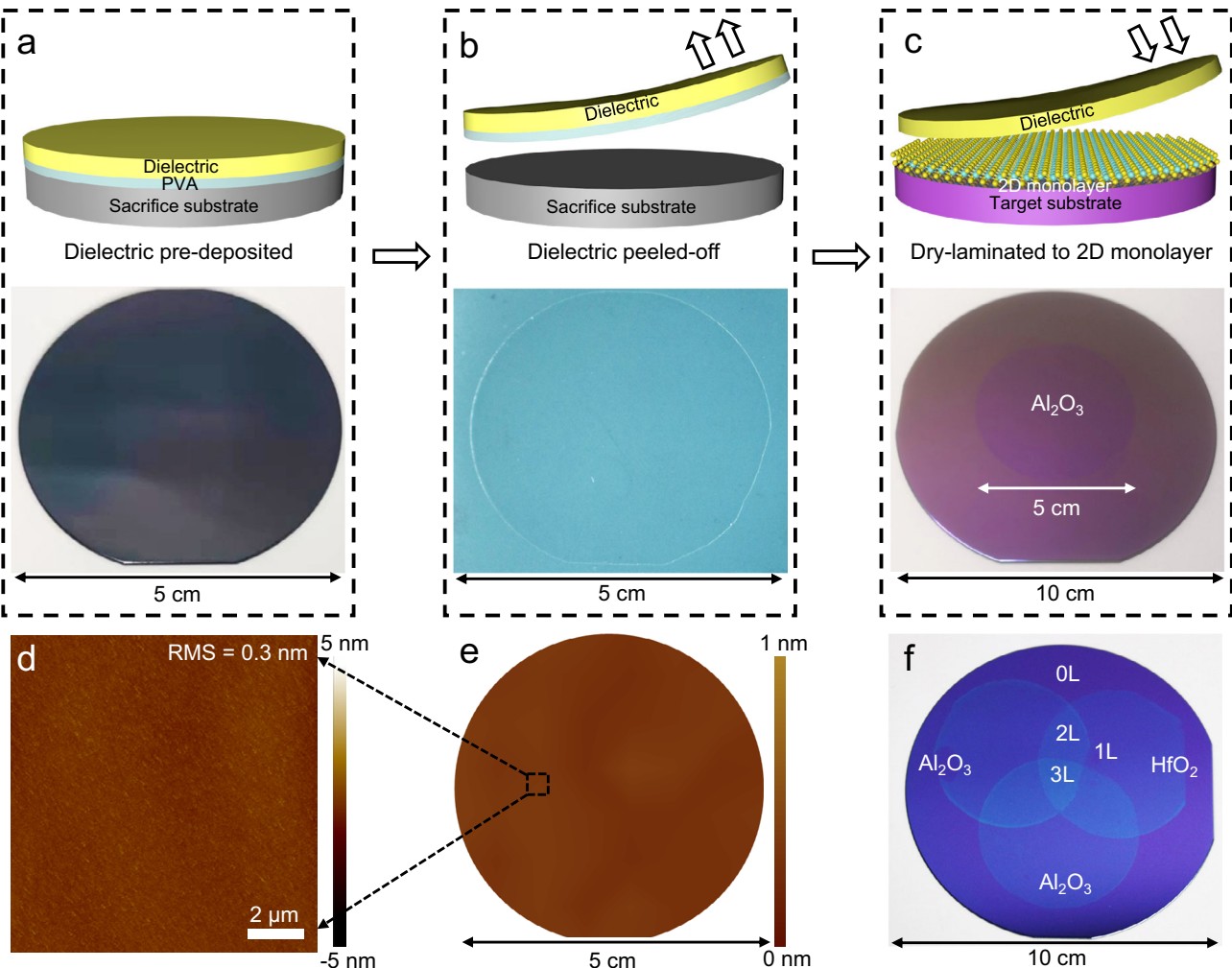

**Fig. 1 | Wafer-scale high-κ dielectric layers transfer process and characterization. a–c** Schematics and optical images of wafer-scale dielectric lamination process with three steps: pre-deposition on PVA sacrifice layer using ALD process (**a**), dielectric dry peeling-off (**b**), and dielectric lamination on top of target substrate (**c**). **d** AFM height measurement for the bottom side of transferred $Al_2O_3$ with a small RMS surface roughness of 0.3 nm, demonstrating atomic scale flat surface. **e** AFM surface roughness mapping of the 2-inch transferred film by measuring the RMS roughness over hundreds of locations across the wafer. **f** Layer-by-layer lamination of 2-inch-size $Al_2O_3$ and $HfO_2$ dielectric film, demonstrating a vdW $Al_2O_3/HfO_2/Al_2O_3$ oxide superlattice in wafer scale.

on glass substrate to avoid parasite capacitance between probing pads, and the effective capacitor size is $20 \times 20\ \mu m^2$. We note the top metal is also vdW laminated within MIM structure using our previous method[28], to avoid the damages of thin dielectrics during the conventional metal evaporation process. The $I–V$ curve of transferred $Al_2O_3$ is measured at room temperature under vacuum environment ($10^{-4}$ Torr). As shown in Fig. 2b, the leakage current remains below $10^{-7}$ A/cm$^2$ for all transferred dielectric before breakdown, which is consistent with the $Al_2O_3$ directly grown on silicon substrate and is 5 orders of magnitude lower than the low-power requirement ($10^{-2}$ A/cm$^2$) from International Technology Roadmap for Semiconductors (ITRS)[29], suggesting the transfer process will not impact the original insulating properties of high-κ dielectric film. Furthermore, the breakdown voltage of $Al_2O_3$ is measured to be 5.4 V, 6.7 V, 8.6 V and 14.3 V for 4.5-nm-thick, 8.3-nm-thick, 11-nm-thick, and 22-nm-thick transferred $Al_2O_3$ film, respectively, corresponding to breakdown electric fields over 6 MV/cm for all measured thickness (Fig. 2c).

Figure 2d plots the $C–V$ measurement of the transferred dielectric film with various thickness under high-frequency of 1 MHz. Overall, the measured capacitance decreases with increasing dielectric thickness, and the dielectric constant could be extracted using following equation: $\varepsilon_{eff} = C_{ox}T_{ox}/\varepsilon_0$, where $\varepsilon_{eff}$, $\varepsilon_0$, $C_{ox}$, $T_{ox}$ are effective dielectric constant, vacuum permittivity, dielectric capacitance, and dielectric thickness, respectively. As shown in Fig. 2e, the $\varepsilon_{eff}$ remains at ~7.4 for

$Al_2O_3$ thickness above 12 nm, closing to the bulk value of $Al_2O_3$. With decreasing thickness, the $\varepsilon_{eff}$ gradually decreases and could be attributed to dielectric dead layers between metal and insulator interface[30–32], where two interfacial capacitances exist at the electrode/dielectric boundaries and act as series-capacitance, leading to reduce dielectric constant with film thickness decrease, consistent with previous study of directly deposited $Al_2O_3$ dielectric[33]. Similarly, the capacitance of transferred $HfO_2$ film is also measured, displaying a similar trend (Fig. 2f and Supplementary Fig. 7). The thinnest thickness for transferred $Al_2O_3$ and $HfO_2$ is 3 and 2.6 nm, respectively, and further reducing the dielectric thickness leads to much increased leakage current, as shown in Supplementary Fig. 8. The highest dielectric capacitance we could achieve is 2.5 μF/cm$^2$ at 1 MHz and 2.8 μF/cm$^2$ at 1 kHz, corresponding to an EOT of 1.38 nm and 1.2 nm, respectively. The relationship between the capacitance and measurement frequency is further plotted in Supplementary Fig. 9. In additional, we also test the thermal stability of vdW dielectric, where consistent capacitance is observed with temperature up to 500 °C, as shown in Supplementary Fig. 10.

## Electrical properties of 2D vdWs top-gate transistors
Within our approach, the dielectric is physically laminated on top of monolayer $MoS_2$ with a weakly-coupled semiconductor-dielectric

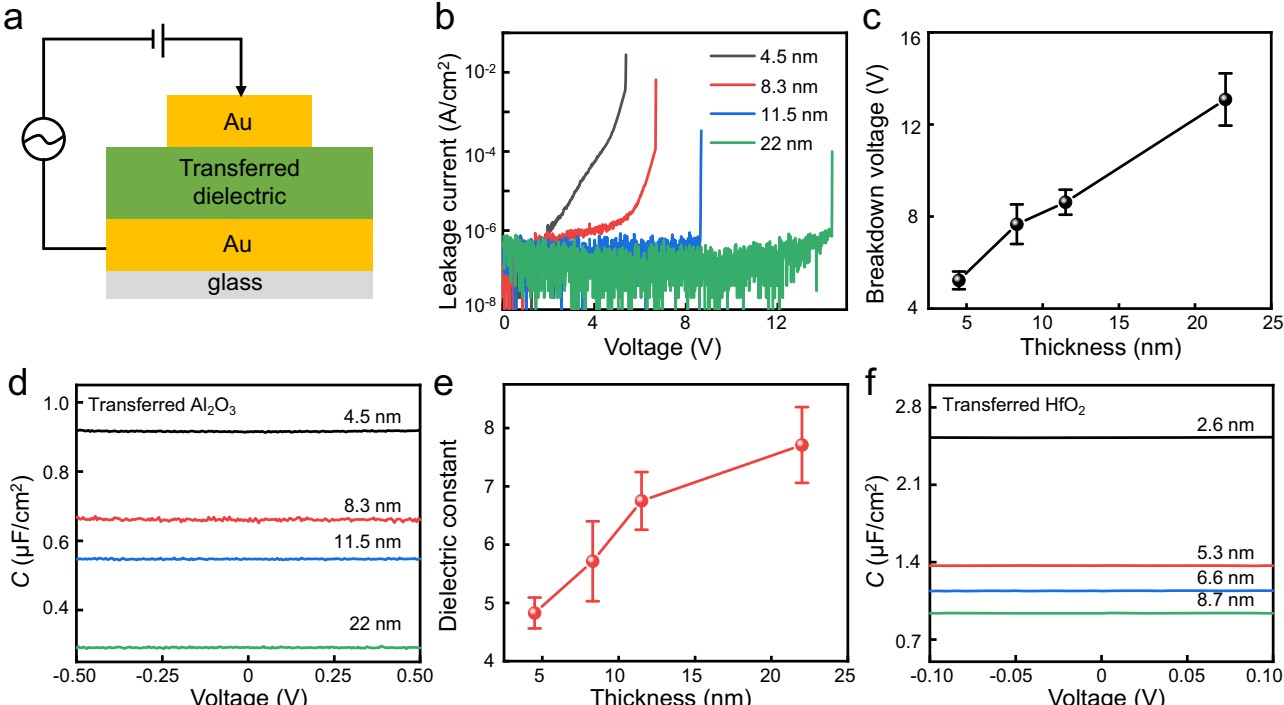

**Fig. 2 | Dielectric properties of transferred oxide films. a** Schematic of metal-insulator-metal (MIM) device structure for capacitance measurement. **b** The leakage current density as a function of applied voltage for transferred $Al_2O_3$ with various thickness. Low leakage current of $10^{-6}$ $A/cm^2$ are observed before dielectric breakdown. **c** Breakdown voltage of the transferred dielectric, where high breakdown electric field over 6 MV/cm are demonstrated. **d**, **e** Capacitance-voltage (C–V) characteristic of transferred $Al_2O_3$ with various thickness at 1 MHz measurement frequency (**d**), and its corresponding dielectric constant. **f** Capacitance-voltage (C–V) characteristic transferred $HfO_2$ (1 MHz frequency) with various thickness, demonstrating highest capacitance of 2.5 $\mu F/cm^2$. The error bars from **c**, **e** are extracted from three devices.

interface, and therefore could maintain the intrinsic properties of delicate monolayer channel. To demonstrate this, back-gated $MoS_2$ transistor arrays are first fabricated using CVD-grown $MoS_2$ monolayer, where Ag/Au (30/20 nm) are thermally deposited as the source-drain electrodes, highly doped silicon and 300 nm thick $SiO_2$ are used back-gate electrode and back-gate dielectric, respectively. The $I_{ds}$-$V_g$ transfer curve and $I_{ds}$-$V_{ds}$ output curve of the back-gated transistor are first measured in a probe station under vacuum ($10^{-4}$ Torr). As shown in Supplementary Fig. 11, the devices exhibit linear output curve with intrinsic n-type behavior, consistent with previous reports for CVD-grown monolayer $MoS_2$ (refs. [34], [35]). Afterwards, two different dielectric integration methods are applied to deposit 10 nm thick $Al_2O_3$ dielectrics on the same transistor array. The first method is using our transfer approach by laminating the pre-fabricated $Al_2O_3$ film, and the other is using conventional seed-assisted growth[24] by thermally depositing 1 nm Al seeds and then directly conducted ALD process on $MoS_2$ surface. As shown in Fig. 3a, the seed-assisted growth method strongly n-dopes and degrades $MoS_2$ for all 50 transistors measured, where the off-state current increases and the on-off ratio decreases 3 orders of magnitudes. In the meantime, the threshold voltage (defined as $I_{ds}$ of 0.1 nA/μm) also shifts to negative direction compared to their intrinsic state (from −13 V to −36 V). This strong doping behavior is consistent with previous literatures using seed-assisted ALD approach, owning to the high-energy seed deposition process and chemical reaction during ALD process[24]. In contrast, for devices with transferred $Al_2O_3$ dielectric, the off-state current and the on-off ratio remains similar to their intrinsic state, as shown Fig. 3b. On the other hand, the threshold voltage also negatively shifts a very small value of 0.8 V after transferred $Al_2O_3$ (Fig. 3c), and could be largely attributed to the change of top-dielectric environment (from vacuum to $Al_2O_3$), rather than the intrinsic defects or dielectric-$MoS_2$ reaction during the integration process. To confirm this, we have mechanically peeled the

transferred $Al_2O_3$ film from $MoS_2$ transistor and measured the transfer curve again. As shown in Supplementary Fig. 12, the threshold voltage restores to their originally state (as-fabricated device without top dielectric) without any doping effect, suggesting the intrinsic $MoS_2$ channel is not impacted during the dielectric integration process. Importantly, the ability to peel the transferred $Al_2O_3$ film is another strong indicator for the weakly interacted dielectric-$MoS_2$ interfaces, in contrast to directly deposited dielectric-2D interfaces that can not be separated once fabricated.

To further confirm the vdW dielectric interface with minimized impact to 2D channel. We have applied the Raman and PL (photoluminescence) to characterize the monolayer $MoS_2$ before and after dielectric integration. As shown in Supplementary Fig. 13, the Raman peaks (E peak and $A_1$ peak) of the monolayer $MoS_2$ remain identical before and after vdW integrating the $Al_2O_3$, indicating non-observable doping effect. Similarly, PL measurement also shows a consistent peak position -1.87 eV before and after integrating $Al_2O_3$, suggesting the minimized charge transfer between dielectric membrane and the underlayer $MoS_2$. Furthermore, we have plotted the PL and Raman mapping data in Supplementary Fig. 13, and uniform mapping results are observed for both samples before and after $Al_2O_3$ integrating.

Without altering the 2D intrinsic properties, our approach could be used to fabricate high-performance top-gate transistors. As shown in Fig. 3d, large-scale top-gate transistor arrays are fabricated by transferring 10 nm $Al_2O_3$ films on $MoS_2$, followed by depositing 10/40 nm Ti/Au as the gate electrodes, and the channel length and width are 5 μm and 2 μm, respectively. Owning to the high-quality transferred top-gate dielectric, high on-off ratio over $10^7$ could be achieved under low operation voltage of 1 V (both $V_g$ and $V_{ds}$), demonstrating lowest subthreshold swing (SS) of 68 mV/dec. We note the operation voltage can be further decreased to 0.5 V by laminating thinner dielectric with

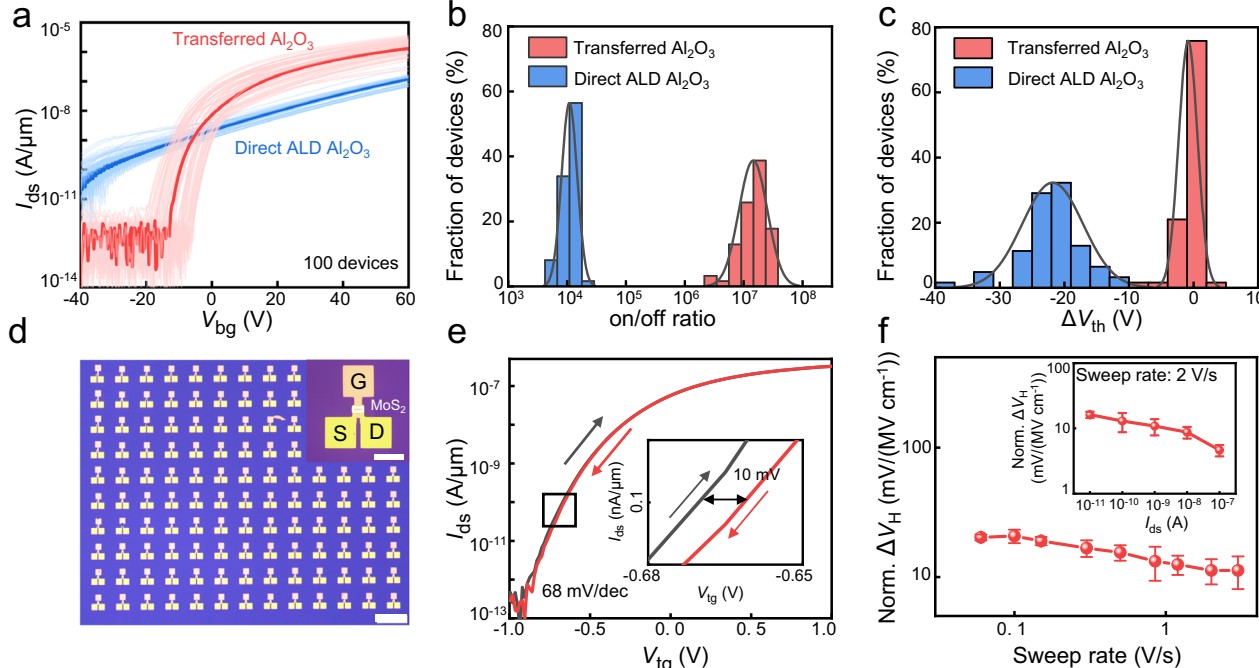

**Fig. 3 | Electrical characterization of MoS$_2$ transistors with transferred dielectrics. a** The $I_{ds}$–$V_g$ transfer characteristic of 100 back-gate MoS$_2$ transistors with both transferred Al$_2$O$_3$ (red lines) or direct ALD Al$_2$O$_3$ (blue lines) with the representative MoS$_2$ transfer cruve highlighted. **b, c** The statistic distribution of on/off ratio (**b**) and threshold voltage shift (**c**) of back-gate MoS$_2$ FETs with transferred Al$_2$O$_3$ and direct ALD Al$_2$O$_3$, respectively, where the solid line is fitted through Gaussian fitting. The devices with transferred dielectric exhibt higher on-off ratio, minimized doping effect. **d** Optical image of top-gate MoS$_2$ transistor array fabricated by transferred Al$_2$O$_3$ dielectric. Scale bar is 200 μm and 50 μm (inset). **e** The double-sweep transfer characteristic of a top-gate MoS$_2$ FET fabricated by transferred Al$_2$O$_3$ as the top-gate dielectric, demonstrating small hysteresis value of 10 mV (inset). **f** The normalized hysteresis (Norm. $\Delta V_H$) as a function of sweeping speed, and as a function of source-drain current (inset). The Norm. $\Delta V_H$ is calculated by using $\Delta V_H/E_g$, where $\Delta V_H$ is the hysteresis and $E_g$ is the gate electric field. The error bars from **f** are extracted from three devices.

smaller EOT (Supplementary Fig. 14), which is desired for low-power operation. Importantly, nearly hysteresis-free switching behavior is also observed, with small hysteresis for all gate sweeping speeds (ranging from 0.06 V/s to 3 V/s, Fig. 3e, f). The lowest hysteresis is 10 mV for top-gate measurement, which is one of lowest value for 2D top-gate dielectric (Supplementary Table S1), further suggesting the clean interface between transferred dielectric and channel materials with minimized defects and interface states ($D_{it}$) using vdW dielectric integration[36]. The $D_{it}$ could be further extracted from 1/$f$ noise approach (Method section and Supplementary Fig. 15), exhibiting small value of 7.6×10$^9$ cm$^{-2}$ eV$^{-1}$, much smaller compared to metal-buffered ALD process[37]. Importantly, our top-gate devices show similar electrical properties and negligible hysteresis after 3 weeks storage as show in Supplementary Fig. 16, indicating the dielectric/semiconductor interface is long-term stable. Our vdW dielectric integration method is not only limited to MoS$_2$, and also could be extended to other 2D semiconductors. To demonstrate this, we have fabricated monolayer WSe$_2$ transistors using Au as the contact metal. As shown in Supplementary Fig. 17, the as-fabricated WSe$_2$ transistor demonstrate p-type $I_{ds}$-$V_{gs}$ transfer curve using 300 nm thick SiO$_2$ as back-gate dielectric. After vdW laminating 10 nm Al$_2$O$_3$ on top of the device, identical device behavior is observed, suggesting the dielectric integration won't impact the intrinsic behavior of WSe$_2$ transistors. Furthermore, we have measured the top-gate WSe$_2$ device properties using vdW Al$_2$O$_3$ as the top-dielectric, and the device exhibits decent high on-off ratio over 10$^6$ and small hysteresis, indicating our method is a universal approach and could be used to fabricate complementary logic circuits.

### Large-scale logic circuits using top-gate MoS$_2$ transistors

The ability to transfer uniform and large-area dielectric films allow us to fabricate more complex devices, such as logic gates and computational circuits. To this end, we constructed monolayer MoS$_2$ (CVD-grown) top-gated transistors arrays using transferred Al$_2$O$_3$ as the top dielectric, where a NMOS logic inverter by connecting two transistors in series. As shown in the inset of Fig. 4a, the gate electrode and source electrode of left transistor (T1) is connected as a pull-up resistor. When a negative $V_{in}$ is applied, the right transistor (T2) is switched-off and the device generates a large $V_{out}$. On the other hand, when a positive $V_{in}$ is applied, the right transistor is switched-on and the $V_{out}$ is zero, leading to function of NMOS inverter[38]. The voltage transfer characteristics of the resulting inverter demonstrates sharp voltage transition with input voltage, yielding a voltage gain of ~50 (Fig. 4a). The demonstrated high voltage gain is essential for the function of signal transmission and logic operation in integrated circuits. We have further constructed more complex logic circuits, NAND, NOR, AND gates by integrating multiple top-gate MoS$_2$ FETs, and achieved the desired logic functions (Fig. 4b–d). The successful realization of these basic logic functions allows us to further construct more complect XOR logic. The XOR gate is integrated by using 4 NAND gates and realizes the corresponding logic function (Fig. 4e, f). The successful construction of the XOR gate show that our method of wafer-scale dielectric lamination can also build the basic computational circuit, further demonstrates the exciting potential for our method in large-area integrated circuits.

### Discussion

In summary, we demonstrate a dry dielectric integration strategy that could transfer wafer-scale Al$_2$O$_3$ and HfO$_2$ on top of 2D semiconductors. The process is compatible with well-developed ALD process and the transferred dielectric film is wafer-scale uniform, demonstrating small equivalent oxide thickness down to 1.2 nm, as well as low leakage current and high breakdown field. Taking the

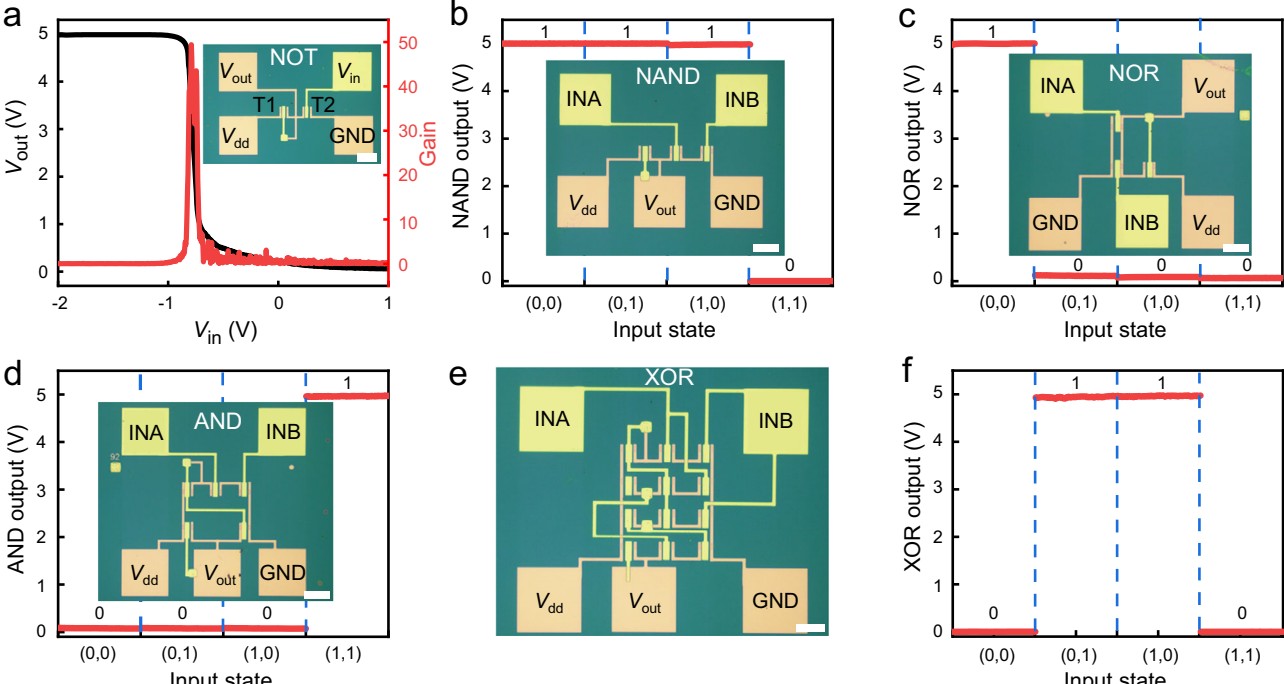

**Fig. 4 | Large-scale logic circuit made by vdWs dielectric integration method.**
**a** The output voltage (black line and left axis) versus input voltage $V_{in}$ and voltage gain (red line and right axis) as a function of $V_{in}$ of a NMOS inverter under $V_{dd}$-5 V. The corresponding optical image is shown as inset. **b**–**d** Output voltage and corresponding optical image (inset) of NAND gate (**b**), NOR gate (**c**) and AND gate (**d**) at for typical input voltage with drain supply voltage at 5 V. **e**, **f** Optical image (**e**) and output-input logic functions (**f**) of a monolayer $MoS_2$ XOR gate. $V_{dd}$, drain supply voltage; $V_{in}$, input voltage; $V_{out}$, output voltage; INA: input A; INB: input B; GND: ground. The scale bars for all-optical images are 50 μm.

advantage of vdW dielectric-semiconductor interface, the intrinsic properties of $MoS_2$ are well-retained during the dielectric integration process, where the top-gate transistor exhibits intrinsic properties with high on-off ratio of $10^7$, small subthreshold swing of 68 mV/dec, and importantly, lowest interface states of $7.6 \times 10^9$ cm$^{-2}$ eV$^{-1}$. We have also constructed functional logic gates and computational circuits, including an NOT, NAND, NOR, AND, and XOR gates. Our study provides a feasible approach to integrate high-κ dielectrics on 2D semiconductors in wafer-scale, and is compatible with standard ALD process with well-controlled thickness and uniformity. This low-energy dielectric integration approach could be further extended to various delicate semiconductors beyond 2D semiconductors (such as perovskite, organic monolayers), which are previously plagued by the ill-defined dielectric-semiconductor interface or are highly prone to degradation during the dielectric deposition process.

## Methods

### Wafer-scale high-κ dielectric layers transfer process

First, PMMA (495 A4, purchased from Kayaku Advanced Materials) is spin-coated (speed 3000 rpm) on $SiO_2$ sacrificial substrate to functionalize the wafer, followed by baking at 150 °C for 2 mins. Next, 9 nm thick PVA layer is spin-coated on top of PMMA. Afterward, atomic layer deposition is conducted using TALD-100 A equipment at 150 °C and this temperature does not destroy the PMMA, and the deposited thickness can be well-controlled through deposition cycles. We used trimethyl aluminum and water as the precursors for $Al_2O_3$ and the tetrakis (dimethylamido) hafnium and water as the precursors for $HfO_2$.

After depositing the dielectrics film, 5 μm PMMA is immediately spin-coated as the support polymer layer, where the PMMA/dielectric/PVA stack could be mechanically peeled-off using thermal release tape or PDMS (polydimethylsiloxane) stamp. The bottom PVA sacrifice layer is dry removed using $O_2$ plasma with 100 W power and the etching time is 90 s. Finally, the PMMA/dielectric is dry-laminated on the target substrate at a temperature of 120 °C and the PMMA support layer is removed using acetone or trichloromethane.

### $MoS_2$ transistors fabrication and electrical measurement

CVD-grown monolayer $MoS_2$ is transferred to highly doped silicon substrate covered with 300 nm thick $SiO_2$. The $MoS_2$ is then patterned into rectangle stripes using conventional photolithography and $O_2$ plasma etching. Next, 30/20 nm Ag/Au electrode pairs are fabricated as source and drain electrode through e-beam lithography and thermal deposition. The electrical measurement is conducted using Agilent B1500A semiconductor analyzer under the vacuum of $10^{-4}$ Torr. The $C-V$ characteristics of different frequencies of MIM devices is conducted using Agilent 4294 A LCR meter under vacuum at $10^{-4}$ Torr.

### Interface states extraction

Interface states $D_{it}$ is extracted by $1/f$ noise method which need to measure the spectral density of top-gate transistor using transferred $Al_2O_3$ film as gate dielectric at different frequency. And the $D_{it}$ could be extracted from the following equation[39]: $\frac{S_I}{I_{ds}^2} = \left(\frac{g_m}{I_{ds}}\right)^2 S_{Vfb}$; $S_{Vfb} = \frac{q^2 K_B T D_{it}}{WLC_{ox}^2 f}$, where $S_I$ is spectral density, $g_m$ is the gate transconductance, $S_{Vfb}$ is the flat band voltage spectral density, $q$ is the electronic charge, $K_B$ is the Boltzmann constant, $T$ is the temperature, $W$ and $L$ are the width and the length of the channel, $C_{ox}$ is the gate capacitance per unit area, respectively.

## Data availability

Relevant data supporting the key findings of this study are available within the article and the Supplementary Information file. All raw data generated during the current study are available from the corresponding authors upon request.

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

## Acknowledgements

Y.L. acknowledge the financial support from the National Key R&D Program of China (no. 2021YFA1200503), the National Natural Science Foundation of China (Grant nos. 51991340, 51991341, 52221001, U22A2074), and the science and technology innovation Program of Hunan Province.

## Author contributions

Y.L. conceived and supervised the research. Y.L. and Zheyi L. designed the experiments. Zheyi L. performed the device fabrication, electrical measurements, and data analysis. Y.C., L.K., Q.T., L.M., D.L., Zhiwei L., W.L., Y. W., and Lei L. contributed to device fabrication. W.D., Liting L., and X.D. contributed to discussions and

data analysis. X.L. contributed to atomic force microscopy measurement and optical characterization. Y.L. and Zheyi L. co-wrote the manuscript. All the authors discussed the results and commented on the manuscript.

## Competing interests

The authors declare no competing interests.
