## [Peer Review File · Nature Communications]

Wafer-scale high- κ dielectrics for two-dimensional circuits via van der Waals integrationEditorial Note: Parts of this Peer Review File have been redacted as indicated to remove third-party material where no permission to publish could be obtained.

REVIEWER COMMENTS

Reviewer #1 (Remarks to the Author):

The submitted paper "Wafer-1 scale high- κ dielectrics for two-dimensional circuits via van der Waals integration," written by Lu et al., demonstrates the realization of 2D semiconductors coupled with high- κ dielectric membranes to achieve superior gate control with a low leakage current. I am sure this work can be vital from society's point of view. The hard limitation in scaling conventional silicon semiconductors can be overcome by replacing semiconductor channels and gate dielectric with new materials of 2D semiconductors and freestanding dielectric membranes. The logic of the experimental approaches is straightforward and detailed.

However, before publication, some questions should be addressed:

1. Please report the area of your devices in all Figures and give all currents as A/ μm or A/ cm^{-2} for better comparability.
2. Please normalize the hysteresis ($\text{mV}/\text{MV cm}^{-1}$) seen in your devices for different sweep rates and bias ranges and compare them to literature values.
3. Raman and PL spectra are useful techniques for evaluating the doping status of 2D TMDCs caused by the environment. I suggest authors provide both Raman and PL mappings to further demonstrate limited doping resulting from top dielectric encapsulation because I still suspect the amorphous nature of ALD-prepared Al_2O_3 or HfO_2 (not dangling-bond-free surface) doesn't lead to charge doping.
4. I noticed that the dielectric constant of HfO_2 layers prepared by the method is far below the standard value. Can authors explain this degradation? Does the growth temperature of ALD HfO_2 limit the final performance? In addition, the upper limit of deposition temperature for this approach is around 200C, which seems unsuitable for emerging high-k or ferroelectric dielectrics whose preparing temperatures are higher. Do authors consider other potential buffer layers?
5. Although an interface state of $7.7 \times 10^{11} \text{ cm}^{-2} \text{ eV}^{-1}$ is on the lower end of reported values for 2D transistors, this is still 2 orders of magnitude larger than Si technology. Any suggestions to improve those? Have the authors tried anneals?

Reviewer #2 (Remarks to the Author):

In this manuscript, Lu et al. exhibit a dry dielectric integration strategy for the high-k top-gated field-effect transistors. This method can realize the wafer-scale transfer of the ultrathin dielectric layer with high uniformity and flatness without cracks. The MoS₂ devices achieve the EOTs of 1.2 nm, a small subthreshold swing down to 68 mV/dec and an ultralow gate leakage current of 10⁻⁷ A/cm² without doping effect. They also used this method to implement the top gate FET array and several logic circuits. Overall, this work is interesting and promising for the scalable integration of electronics based on 2D materials. These results are expected to deliver a significant impact to the community. In the meantime, some serious issues still need to be addressed carefully before considering this work. My comments are listed as the following.

1. The characterization of the transfer process is very weak, and the authors only show the wafer-level macroscopic morphology and schematic diagram. It is essential to provide more details at every step. Such as HR TEM, EDS Mapping, XPS, Raman/PL.... How is Fig.1f obtained, using AFM? It is basically impossible to do so smoothly. The author may need to redo it with a stair meter and provide a 3D graph of the AFM results. In addition, regarding the transfer process, it is worthwhile to provide a complete video.
2. The authors claim that the wafer-level dielectric films do not experience any cracks during the transfer process. How do you know this? The authors need to provide more sufficient experimental proof for it.
3. In the device fabrication process, how does the transfer of the wafer-level dielectric layer face

the undulating surface where the S/D electrodes have been constructed? Since the oxide dielectric is a rigid material, it is easily destroyed when it is squeezed strongly. How does the author evaluate the application potential of this technology in future advanced manufacturing processes?

4. Many important developments have been made in gate dielectric integration on 2D electronic devices in recent years, such as Nature Electronics 4, 906–913 (2021); Nature Electronics 5, 643–649 (2022); Nature 605, 262–267 (2022); Nature Electronics, 2, 563–571(2019). Nature Electronics 2, 230–235 (2019) and so on. It is important to conduct a systematic and comprehensive comparison and analysis to highlight the uniqueness of this work or solve the unresolved pain points.

5. How is the long-term stability of the device's performance? Especially whether it can tolerate the back-end integration process temperature ($>450^{\circ}\text{C}$).

6. The authors claim that this method has no doping effect on 2D materials, which is critical for CMOS integration. Is this approach universal? Is it also applicable to P-type 2D materials? If it is universal, it is very meaningful that the logic circuit of CMOS is the same as that in Figure

7. Why does the gate dielectric prepared by this method have such a low leakage current? But its dielectric constant is much smaller than the theoretical value, especially HfO_2 , which means that the dielectric has poor dielectric properties. In the ALD process, using a polymer as substrate, won't the polymer have a bad effect on the oxide? It is mentioned in the method that before spin-coating PVA, a layer of PMMA needs to be spin-coated. Will the process temperature of 150°C destroy PMMA? As far as I know, the glass transition temperature of PMMA is 125°C .

Reviewer #3 (Remarks to the Author):

In this article, Lu et al. demonstrate a van der Waals integration method to transfer large-area, high-k dielectrics on 2D semiconductors. They achieve this by cautiously selecting the sacrificial layer, i.e. PVA, which act as both a buffer layer for ALD deposition of Al_2O_3 and HfO_2 , and a protection layer during transferring. Based on such integration, high-performance MoS_2 field-effect transistors can be realized, which exhibit negligible residual doping, a small subthreshold swing, and operation under low operation voltages. Furthermore, a variety of logic gates are realized by exploiting these MoS_2 field-effect transistors. The results are of high quality and should lead to a significant advancement in the 2D-dielectric integration technology. Thus, I recommend its publication after the following questions being addressed.

1. The dielectric constant of transferred HfO_2 is significantly lower than the theoretical value. What could be the reason? Is it due to the transfer process or ALD growth conditions?

2. The thinner Al_2O_3 also display very small leakage current, but the authors chose 10-nm-thick Al_2O_3 for the fabrication of MoS_2 transistors and logic gates. The rationale behind this choice should be provided.

3. Besides PVA, PMMA, and PPC, has the authors considered other sacrificial polymer layer? Is there a general selection rule?

4. Have the authors tried to apply this integration method to other 2D materials? If so, is it reproducible?

5. The connection and working mechanism of the NMOS logic inverter can be described in more detail.

6. The authors have measured quite a few devices. So error bars could be added to Fig. 2c, 2e and Fig. 3f to make them more sound.

Response to Reviewer #1

General comment: The submitted paper "Wafer-1 scale high-k dielectrics for two-dimensional circuits via van der Waals integration," written by Lu et al., demonstrates the realization of 2D semiconductors coupled with high-k dielectric membranes to achieve superior gate control with a low leakage current. I am sure this work can be vital from society's point of view. The hard limitation in scaling conventional silicon semiconductors can be overcome by replacing semiconductor channels and gate dielectric with new materials of 2D semiconductors and freestanding dielectric membranes. The logic of the experimental approaches is straightforward and detailed.

However, before publication, some questions should be addressed:

Response: We thank reviewer for positive comment "I am sure this work can be vital from society's point of view", and "The hard limitation in scaling conventional silicon semiconductors can be overcome", and support for its publication. We also appreciate the specific questions raised and would like to take this opportunity to further clarify these questions below.

Specific Comment 1. Please report the area of your devices in all Figures and give all currents as A/ μm or A/ cm^2 for better comparability.

Response: We thank the reviewer for this important suggestion, and we have now included the devices area and consistently used the units of A/ μm and A/ cm^2 for all measured currents for better comparability, in the revised manuscript.

Specific Comment 2. Please normalize the hysteresis ($\text{mV}/\text{MV cm}^{-1}$) seen in your devices for different sweep rates and bias ranges and compare them to literature values.

Response: We thank the reviewer for this important suggestion. Taken the reviewer suggestion, we now normalize the hysteresis by using the gate electric field V_g/d_{ox} [*Nat. Electron.* 2, 230 (2019)], where the d_{ox} is the thickness of oxide films and the V_g is the gate voltage. As shown in Fig. R1a, b below, the normalized hysteresis is plotted for different sweep rates and bias ranges. Furthermore, we have also compared our normalized hysteresis to other literatures (Fig. R1c) [*Nature* 605, 262 (2022). *Nat. Electron.* 5, 643 (2022); *Nat. Electron.* 2, 563 (2019); *AIP Adv.* 5, 057102 (2015); *IEEE Electr. Device L.* 38, 1763 (2017); *Appl. Phys. Express* 9, 095202 (2016)], where lower hysteresis is observed in our device due to the pristine vdW interface.

We thank the reviewer for this question, and we have included the normalized hysteresis in the revised manuscript.

Fig. R1. a, b, Normalized hysteresis as function of sweeping speed (a) and different gate voltage range (b). **c,** Comparison of normalized hysteresis with other literatures.

Specific Comment 3. Raman and PL spectra are useful techniques for evaluating the doping status of 2D TMDCs caused by the environment. I suggest authors provide both Raman and PL mappings to further demonstrate limited doping resulting from top dielectric encapsulation because I still suspect the amorphous nature of ALD-prepared Al₂O₃ or HfO₂ (not dangling-bond-free surface) doesn't lead to charge doping.

Response: We thank the reviewer for this important question. Following the reviewer's suggestion, we have conducted both Raman and PL characterization on MoS₂ monolayer (Fig. R2). To provide a fair comparison, Raman and PL mappings are conducted on both pristine MoS₂ (before Al₂O₃ lamination) as well as the same MoS₂ location after vdW integrating Al₂O₃ dielectric. As shown in Fig. R2d, the Raman peaks (*E* peak and *A*₁ peak) of MoS₂ remain identical before and after vdW integrating the wafer-scale Al₂O₃, indicating non-observable doping effect. Similarly, PL measurement also shows a consistent peak position ~1.87 eV before and after integrating Al₂O₃ (Fig. R2h), suggesting the minimized charge transfer (or doping effect) between dielectric membrane and the underlayer MoS₂. Furthermore, we have plotted the PL and Raman mapping data in Fig. R2b, c, f, g, and uniform mapping results are observed for both samples before and after Al₂O₃ lamination, further highlighting the uniformity of vdW integrating process.

We thank the reviewer for this important question, and we have included the PL and Raman mapping data in the revised manuscript.

Fig. R2. **a**, Optical image of monolayer MoS₂ as-exfoliated on SiO₂ substrate. **b**, **c**, Raman mapping (A₁' peak position) and PL mapping of the monolayer MoS₂. **d**, The Raman spectra of monolayer MoS₂ before and after transferred Al₂O₃ dielectric. **e**, Optical image of monolayer MoS₂ after integrating Al₂O₃ dielectric. **f**, **g**, Raman mapping (A₁' peak position) and PL mapping of the monolayer MoS₂. **h**, The Raman spectra of monolayer MoS₂ before and after transferred Al₂O₃ dielectric.

Specific Comment 4. I noticed that the dielectric constant of HfO₂ layers prepared by the method is far below the standard value. Can authors explain this degradation? Does the growth temperature of ALD HfO₂ limit the final performance? In addition, the upper limit of deposition temperature for this approach is around 200 °C, which seems unsuitable for emerging high-k or ferroelectric dielectrics whose preparing temperatures are higher. Do authors consider other potential buffer layers?

Response: We thank the reviewer for this important question about HfO₂ dielectric constant. First of all, we found the dielectric constant (κ) of our HfO₂ film decrease with film thickness, and the κ value drops to ~ 10 with sub-10 nm film. This behavior is consistent with previous literatures, where relatively low dielectric constant ($\kappa=7-10$) is measured with similar sub-10 nm HfO₂ film [*Appl. Phys. Lett.* 101, 172910 (2012); *J. Electrochem. Soc.* 151, F189 (2004); *Thin Solid Films* 491, 328 (2005)], as shown in Fig. R3a below. This reduced- κ value and thickness-dependence- κ could be explained by the dielectric dead layer effect [*Nature* 443, 679 (2006); *Appl. Phys. Lett.* 101, 172910 (2012); *Nature* 605, 262 (2022)], where two interfacial capacitances (C_{i1} and C_{i2}) exist at the electrode/dielectric boundaries, as schematical illustrated in Fig. R3b. These interfacial capacitances typically exhibit lower κ -value (hence termed dead layers) and act as series-capacitance with the pristine bulk capacitance (C_{bulk}). Therefore, the overall dielectric constant is smaller than its bulk value, and such effect is more and more pronounced with decreasing film thickness. **To confirm this theory, we have conducted additional control experiment by vdW laminating thicker HfO₂ (25 nm thick). As shown in Fig. R3c, the measured dielectric constant is increased to 15, much closer to its bulk value (~ 20).**

Furthermore, following the reviewer suggestions, we have also measured the dielectric constant of HfO₂ with different growth temperature. As shown in Fig. R3d, relatively low- κ value is consistently observed at all growth temperature, further indicating the dielectric performance is not limited by the growth temperature, but is more governed by the dead layer effect, as explained above.

Finally, regarding to other emerging high- κ or ferroelectric dielectrics, we fully agree with the reviewer that their preparing temperatures could be higher and may not be compatible with our polymer buffer layer. Therefore, other layered buffer layer may be desired (such as graphene), which could also be easily separated from substrate for the lamination of dielectric film. However, 2D buffer do not have dangling bonds and seed layer need to be deposited first prior to the ALD process, limiting the achievement of ultra-thin dielectric film. In additional, the scalability of 2D buffer (by CVD) may not be as good as the polymer buffer (by spin-coating).

We thank the reviewer for these questions, and we have explained the low- κ value of HfO₂ and discussed potential buffers for other emerging dielectric in the revised manuscript.

[redacted]

Fig. R3. **a**, The EOT and dielectric constant value of HfO₂ versus its thickness. This figure is replotted from [*J. Electrochem. Soc.* 151, F189 (2004)]. **b**, Schematic of dielectric dead layer effect, which is replotted from [*Nature* 605, 262-267 (2022)]. **c**, Dielectric constant of HfO₂ with different thickness, for both direct grown HfO₂ (red line), as well as for vdW laminated HfO₂ film (blue line). **d**, Dielectric constant of HfO₂ film as a function of their grown temperature.

Specific Comment 5. Although an interface state of $7.7 \times 10^{11} \text{ cm}^{-1} \text{ eV}^{-1}$ is on the lower end of reported values for 2D transistors, this is still 2 orders of magnitude larger than Si technology. Any suggestions to improve those? Have the authors tried anneals?

Response: We thank the reviewer for this important question regarding to the interface state (D_{it}). In general, two major approaches are used for extracting D_{it} and its value is strongly depended on the extraction technique. The first method (also used in our

manuscript) is subthreshold (SS) approach based on the following equations ($SS = \ln(10) \frac{K_B T}{q} (1 + \frac{q D_{it}}{C_{ox}})$), where the D_{it} can be extracted by measuring the transistor SS value. However, this method is largely limited by the theoretical SS limit (60 mV/dec) and the possible influence of Schottky barrier. Hence, relatively high D_{it} on the order of $10^{11} \text{ cm}^{-2} \text{ eV}^{-1}$ is achieved even in Si based devices [*ECS Solid State Lett.* 2, Q32 (2013)], similar to the measured D_{it} in our manuscript.

The second method is based on $1/f$ noise measurement, where the D_{it} could be extracted from the following equation [*Nanoscale* 6, 433 (2014)]:

$$\frac{S_I}{I_{ds}^2} = \left(\frac{g_m}{I_{ds}}\right)^2 S_{Vfb}$$

$$S_{Vfb} = \frac{q^2 K_B T D_{it}}{W L C_{ox}^2 f}$$

where S_I is spectral density, g_m is the gate transconductance, S_{Vfb} is the flat band voltage spectral density, q is the electronic charge, K_B is the Boltzmann constant, T is the temperature, W and L are the width and the length of the channel, C_{ox} is the gate capacitance per unit area, respectively. To extract the D_{it} based on $1/f$ method, we have measured the S_I as a function of frequency for our MoS_2 device with vdW dielectric. As shown in Fig. R4, the extracted D_{it} using this method is $7.6 \times 10^9 \text{ cm}^{-2} \text{ eV}^{-1}$, which is around two orders of magnitude lower than the SS extraction method. At the same time, our value is also close to D_{it} of Si device using this method, which is on the order of $10^9 \text{ cm}^{-2} \text{ eV}^{-1}$ [*Microel. Reliab. R.* 37, 1599 (1997)], as also noted by the reviewer. Hence, the observed large D_{it} (compared to Si device) could be largely attributed to the different extraction method (SS in our method vs. $1/f$ for Si device), and have less relationship with the interface quality and the annealing effect.

We thank the reviewer for these questions, and we have further included the D_{it} using $1/f$ extraction method in the revised manuscript.

Fig. R4. Noise spectra as a function of frequency with different gate bias at $V_{ds} = 1 \text{ V}$.

Response to Reviewer #2

General comment: In this manuscript, Lu et al. exhibit a dry dielectric integration strategy for the high-k top-gated field-effect transistors. This method can realize the wafer-scale transfer of the ultrathin dielectric layer with high uniformity and flatness without cracks. The MoS₂ devices achieve the EOTs of 1.2 nm, a small subthreshold swing down to 68 mV/dec and an ultralow gate leakage current of 10⁻⁷A/cm² without doping effect. They also used this method to implement the top gate FET array and several logic circuits. Overall, this work is interesting and promising for the scalable integration of electronics based on 2D materials. These results are expected to deliver a significant impact to the community. In the meantime, some serious issues still need to be addressed carefully before considering this work. My comments are listed as the following.

Response: We thank reviewer for carefully reading our manuscript and the recognition that “this work is interesting and promising for the scalable integration of electronics based on 2D materials”, and “These results are expected to deliver a significant impact to the community”. We particularly appreciate the highly insightful and constructive comments from the reviewer and welcome the opportunity to address these questions and describe the revisions we have made accordingly.

Specific Comment 1. The characterization of the transfer process is very weak, and the authors only show the wafer-level macroscopic morphology and schematic diagram. It is essential to provide more details at every step. Such as HR TEM, EDS Mapping, XPS, Raman/PL.... How is Fig.1f obtained, using AFM? It is basically impossible to do so smoothly. The author may need to redo it with a stair meter and provide a 3D graph of the AFM results. In addition, regarding the transfer process, it is worthwhile to provide a complete video.

Response: We thank the reviewer for these insightful questions regarding to the film quality. First of all, we would like to clarify the measurement method of Fig. 1e (also shown as Fig. R5a below). The Fig. 1e is obtained by measuring over 100 different locations across a 2-inch size Al₂O₃ using AFM, and each location have a scanning area of 10 μm×10 μm (with a spacing of ~2 mm between two adjacent locations). The root mean square (RMS) roughness of each location could be obtained by analyzing the corresponding AFM height profile. Next, the RMS roughness of 100 different locations were plotted in a two-dimensional graph, leading to the generation of Fig. 1e.

Next, we also thank the reviewer suggestion that “The author may need to redo it with a stair meter and provide a 3D graph of the AFM results”. Taken this suggestion, we have plotted the 3D AFM height profile of one measurement location, as shown in Fig. R5b. Since the AFM can only measure the height profile in limited area, stair meter is further applied to measure the surface morphology of vdW laminated Al₂O₃. As shown in Fig. R5c, the stair meter also demonstrates a flat surface without any clear steps, and the measured roughness is 0.98 nm across millimeter length. We note the observed larger roughness of stair meter (compared to AFM) could be attributed to its intrinsic low resolution in vertical direction.

Fig. R5. **a**, Replot of Fig. 1e from main manuscript. **b**, 3D AFM graph of one location of freestanding dielectric film. **c**, One-dimensional height mapping of the Al_2O_3 film using stair-meter for three different locations, demonstrating flat surface across larger area.

Furthermore, we have also followed the reviewer suggestion and applied XPS and Raman/PL mapping at every step during our fabrication process. As shown in Fig. R6, the XPS peaks of Al_2O_3 dielectric remain identical during different fabrication processes, including dielectric pre-deposited step (Fig. R6a), dielectric peeled-off step (Fig. R6b), as well as dielectric lamination step (Fig. R6c), indicating our process could maintain the properties of dielectric film. Similarly, both the Raman and PL peaks of MoS_2 remain identical before and after vdW integrating the wafer-scale Al_2O_3 , indicating the minimized impact between dielectric membrane and the underlayer MoS_2 (Fig. R6d-k). Furthermore, we have plotted the PL and Raman mapping data in Fig. R6e, f, i, j, and uniform mapping results are observed for both samples before and after Al_2O_3 integrating, further highlighting the uniformity of vdW lamination.

Fig. R6. **a-c**, The XPS spectra of Al_2O_3 at different fabrication steps, including dielectric pre-deposition step (**a**), dielectric peeled-off step (**b**), as well as dielectric lamination step (**c**). **d**, Optical image of monolayer MoS_2 as-exfoliated on SiO_2 substrate. **e, f**, Raman mapping (A_1' peak position) and PL mapping of the monolayer MoS_2 . **g**, The Raman spectra of monolayer MoS_2 before and after transferred Al_2O_3 dielectric. **h**, Optical image of monolayer MoS_2 after integrating Al_2O_3 dielectric. **i, j**, Raman mapping (A_1' peak position) and PL mapping of the monolayer MoS_2 after dielectric lamination. **k**, The PL spectra of monolayer MoS_2 before and after transferred Al_2O_3 dielectric.

Finally, we have also followed the reviewer suggestion and taken a video of the dielectric film transfer and lamination process, as shown in the uploaded Movie #1, and some screenshots of this video are shown in Fig. R7 below.

We thank the reviewer for these important questions, and we have further clarified the measurement method of Fig. 1e, included the stair meter, PL/Raman mapping and XPS data, and added the process video in the revised manuscript.

Fig. R7. **a-d**, Screenshots of the uploaded video for some key steps during the transfer process, including dielectric peeling-off (**a**), buffer layer etching (**b**), dielectric lamination (**c**), and the final sample (**d**).

Specific Comment 2. The authors claim that the wafer-level dielectric films do not experience any cracks during the transfer process. How do you know this? The authors need to provide more sufficient experimental proof for it.

Response: We thank the reviewer for the question about the film quality, and the crack of dielectric film (both Al_2O_3 or HfO_2) film can be directly observed and confirmed under optical microscope. To demonstrate this, we have fabricated another control sample with cracks by intentionally introducing strains during the film transfer process. As shown in Fig. R8a, cracks can be clearly identified in optical images, due to the optical contrast between substrate and 10 nm thick Al_2O_3 . In great contrast, for devices using our lamination technique with minimized strains, we did not observe any cracks under optical microscope across the 2-inch wafer. As shown in Fig. R8b, we have included 10 representative images (each size of $1.2 \text{ mm} \times 1 \text{ mm}$) at different locations, and uniform film are consistently observed, indicating the transferred film is continuous without cracks.

Furthermore, the absence of film crack could be confirmed through the height profile measurement. As discussed in previous response (to comment #1), the film is continuous and flat with small RMS roughness (0.3 nm) under AFM measurement, for over 100 different locations across a 2-inch wafer. At the same time, the stair meter results also demonstrate flat surface without any clear steps across larger area ($\sim 0.5 \text{ mm}$), indicating the film is continuous without cracks.

We thank the reviewer for this question, and we have further discussed the crack-free film in the revised manuscript.

Fig. R8. a, Optical images of control sample by intentionally introducing large strain during lamination process, where the cracks can be clearly observed. **b**, Optical images of 10 different locations of crack-free Al_2O_3 film, demonstrating flat surface without clear cracks.

Specific Comment 3. In the device fabrication process, how does the transfer of the wafer-level dielectric layer face the undulating surface where the S/D electrodes have been constructed? Since the oxide dielectric is a rigid material, it is easily destroyed when it is

squeezed strongly. How does the author evaluate the application potential of this technology in future advanced manufacturing processes?

Response: We thank the reviewer for this question. To investigate the interfaces between raised electrode and the flat dielectric, we have conducted AFM measurement around the electrode edge region. For as-fabricated electrodes (50 nm thick Au), sharp electrode edges are observed with a clear height step of 50 nm, as shown in Fig. R9a-c. In contrast, for electrode covered with 10 nm thick Al₂O₃ film using our vdW lamination process, obvious slope is observed at the electrode edge region, as shown in Fig. R9d-f. This observation indicates the transferred dielectric film is not fully contacted with the electrode sidewalls and there is a vacuum gap in between, as schematically illustrated in Fig. R9d.

As the reviewer pointing out, this vacuum gap could pose challenges in future advanced manufacturing processes, especially for ultra-scaled devices. To overcome this limitation, two possible pathways could be investigated in the future. The first method is reducing the thickness of raised electrodes or using buried electrodes (schematically illustrated in Fig. R9g), which could reduce the electrode edge steps height and forms an intimate contact with the vdW dielectric. The second method is using pre-patterned dielectric and then laminate them onto 2D surface, where the source-drain electrode could be deposited through the patterns (window) of dielectric, as schematically illustrated in Fig. R9h. Within this approach, the source-drain electrode is fabricated after laminating the dielectric, hence the electrode edge won't impact their interface with the dielectric.

Fig. R9. a-c, AFM measurement of as-fabricated electrode edge, exhibiting sharp edge with a height of 50 nm. d-f, AFM measurement of the electrode edge covered with vdW dielectric, demonstrating a clear height slope within ~500 nm length region. g, h, Two possible pathways for further reducing the edge effect of electrode, including the buried electrode (g) as well as the patterned dielectric transfer (h).

Specific Comment 4. Many important developments have been made in gate dielectric integration on 2D electronic devices in recent years, such as Nature Electronics 4, 906-913 (2021); Nature Electronics 5, 643-649 (2022); Nature 605, 262-267 (2022); Nature Electronics, 2, 563-571(2019). Nature Electronics 2, 230-235 (2019) and so on. It is important to conduct a systematic and comprehensive comparison and

analysis to highlight the uniqueness of this work or solve the unresolved pain points.

Response: We thank the reviewer for this question and we have now compared our key results with the mentioned references by reviewer and other related methods for dielectric integration on 2D semiconductors [*Nat. Electron.* 2, 230 (2019); *Nature* 605, 262 (2022); *Nat. Electron.* 5, 643 (2022); *Nat. Electron.* 2, 563 (2019); *AIP Adv.* 5, 057102 (2015); *Appl. Phys. Express* 9, 095202 (2016); *Jpn. J. Appl. Phys.* 60, SBBH03 (2021). Ref. 1-8 in Table 1]. As shown in Table R1 below, our vdW lamination method exhibits high scalability due to the use of polymer buffer and the well-controlled ALD process, as well as small hysteresis and SS values due to vdW interfaces with minimized interfacial states. More importantly, we could achieve lower EOT value using well-developed high- κ dielectric such as Al₂O₃ and HfO₂, further highlighting its potential for practical application.

We thank the reviewer for this question, and we have included the Table R1 in the revised manuscript to further highlight the uniqueness of our work.

Table R1. Comparison of our vdW process with other dielectric integration method.

Dielectric	Method	scalability	Hysteresis (mV)	EOT (nm)	SS (mV/dec)	Ref.
CaF ₂	CVD	Micrometer	35	0.9	93	1
SrTiO ₃	transfer	Centimeter	42	1	71	2
β -Bi ₂ SeO ₅	Plasma oxidation	Micrometer	60	0.5	62	3
PTCDA/HfO ₂	CVD/ALD	Large-area	10	2	64	4
Al ₂ O ₃	ALD	Wafer-scale	452	23	138	5
HfO ₂	ALD	Wafer-scale	125	10.5	233	6
Sb ₂ O ₃	Thermal evaporation	Wafer-scale	10	3.3	68	7
ZrO ₂	ALD	Wafer-scale	155	2.3	90	8
Al ₂ O ₃ or HfO ₂	transfer	Wafer-scale	10	1.3	68	This Work

Specific Comment 5. How is the long-term stability of the device's performance? Especially whether it can tolerate the back-end integration process temperature (>450°C).

Response: Thanks for these important questions about the device long-term stability and temperature stability. First, to test the time-related stability, we have measured the I_{ds} - V_{gs} transfer curve of the MoS₂ transistor using vdW Al₂O₃ as the top gate dielectric (Fig. R10a). After three weeks storage at ambient environment, the device exhibits similar electrical properties compared to the as-fabricated device (Fig. R10b), suggesting the decent stability of our device. Importantly, both devices show negligible hysteresis, indicating the dielectric interface is also stable after three weeks storage.

Furthermore, according to the reviewer suggestion, we have also measured the temperature stability of our vdW Al₂O₃ film. The device structure is schematically

illustrated in Fig. R10c (inset), where the vdW Al_2O_3 film is sandwiched between two metal electrodes. As shown in Fig. R10c, d, the resulting device demonstrate consistent capacitance with temperature increased to $500\text{ }^\circ\text{C}$, indicating the vdW dielectric is compatible with back-end-of-line (BEOL) process temperature ($450\text{ }^\circ\text{C}$).

We thank the reviewer for this question, and we have discussed the time and thermal stability of our vdW dielectric in the revised manuscript.

Fig. R10. **a**, I_{ds} - V_{tg} transfer curves of as-fabricated MoS_2 transistor. **b**, I_{ds} - V_{tg} transfer curves of MoS_2 transistor measured after 3 weeks storage, demonstrating negligible change. **c**, C-V measurement of as-fabricated vdW Al_2O_3 films. **d**, C-V measurement vdW Al_2O_3 films after $500\text{ }^\circ\text{C}$ annealing.

Specific Comment 6. The authors claim that this method has no doping effect on 2D materials, which is critical for CMOS integration. Is this approach universal? Is it also applicable to P-type 2D materials? If it is universal, it is very meaningful that the logic circuit of CMOS is the same as that in Figure 4.

Response: Thanks for the excellent point. Within our method, the wafer-scale dielectric is mechanically laminated on top of the 2D surface and is not limited by the type of the 2D channel. Hence, it is a universal approach and could be extended to other p-type 2D semiconductors. To demonstrate this, we have fabricated monolayer WSe_2 transistors using our approach. As shown in Fig. R11a, the as-fabricate WSe_2 transistor demonstrate p-type I_{ds} - V_{gs} transfer curve using 300 nm thick SiO_2 as back-gate dielectric. After vdW laminating 10 nm Al_2O_3 on top of the device, identical device behavior is observed, suggesting the dielectric integration won't impact the intrinsic behavior of p-type WSe_2 transistors. Furthermore, we have measured the top-gate device properties using vdW Al_2O_3 as the top-dielectric, and the device exhibits decent p-type behavior with high on-off ratio over 10^6 (Fig. R11b), indicating our method is a universal approach and could be used to construct wafer-scale CMOS logic for 2D semiconductors.

We thank the reviewer for this question, and we have included the p-type device performance of vdW WSe_2 transistors and discussed its potential for CMOS application in the revised manuscript.

Fig. R11. a, I_{ds} - V_{gs} transfer curves of WSe₂ transistors before and after vdW integrating Al₂O₃ dielectric. b, I_{ds} - V_{tg} transfer curves of top-gate WSe₂ transistor fabricated using vdW dielectric.

Specific Comment 7. Why does the gate dielectric prepared by this method have such a low leakage current? But its dielectric constant is much smaller than the theoretical value, especially HfO₂, which means that the dielectric has poor dielectric properties. In the ALD process, using a polymer as substrate, won't the polymer have a bad effect on the oxide? It is mentioned in the method that before spin-coating PVA, a layer of PMMA needs to be spin-coated. Will the process temperature of 150 °C destroy PMMA? As far as I know, the glass transition temperature of PMMA is 125 °C.

Response: We thank the reviewer for this important question about the film quality. First of all, we note the dielectric constant (κ) of our HfO₂ decreases with its thickness, dropping to a low κ value ~ 10 (with sub-10 nm thickness) and is much smaller than its theoretical value (~ 20). This behavior is consistent with previous literatures, where relatively low dielectric constant ($\kappa=7-10$) is also measured with similar sub-10 nm HfO₂ film [*Appl. Phys. Lett.* 101, 172910 (2012); *J. Electrochem. Soc.* 151, F189 (2004); *Thin Solid Films* 491, 328 (2005).], as shown in Fig. R12a below. This reduced- κ value and thickness-dependence- κ could be explained by the dielectric dead layers [*Nature* 443, 679 (2006); *Appl. Phys. Lett.* 101, 172910 (2012); *Nature* 605, 262 (2022)], where two interfacial capacitances (C_{i1} and C_{i2}) exist at the electrode/dielectric boundaries, as schematical illustrated in Fig. R12b. These interfacial capacitances typically exhibit lower κ -values (hence termed dead layers) and act as series-capacitance with the pristine bulk capacitance (C_{bulk}). Hence, the overall dielectric constant is smaller than its bulk value, and such effect is more and more pronounced with decreasing film thickness. To confirm this theory, we have conducted additional control experiment by vdW laminating thicker HfO₂ (25 nm thick). As shown in Fig. R12c, the measured dielectric constant is ~ 15 , much closer to its bulk value (~ 20). Based on the above analysis, we believe the reduced- κ value is more limited by the dead layer effect, rather than by intrinsic quality. This is also supported by the low leakage current and high device performance of vdW film, as pointed out by the reviewer.

Furthermore, regarding to the temperature impact, we note the 150 °C processing won't destroy the PMMA. In fact, the PMMA (purchased from Kayaku Advanced Materials) is widely used as e-beam resist for nano-device fabrication, and its standard recipe includes 180 °C baking after PMMA solution spin-coating (as copied in Fig. R12d below), indicating the PMMA is not damaged during lower processing temperature (150 °C).

We thank the reviewer for these questions, and we have explained the low- κ value of HfO₂ and discussed temperature impact to PMMA in the revised manuscript.

[redacted]

Fig. R12. **a**, The EOT and dielectric constant value of HfO₂ versus its thickness. This figure is replotted from [*J. Electrochem. Soc.* 151, F189 (2004)]. **b**, Schematic of dielectric dead layer effect, which is replotted from [*Nature* 605, 262-267 (2022)]. **c**, Dielectric constant of HfO₂ with different thickness, for both direct grown HfO₂ (red line), as well as for vdW laminated HfO₂ film (blue line). **d**, The standard recipe of PMMA from Kayaku Advanced Materials. [kayakuam.com].

Response to Reviewer #3

General comment: In this article, Lu et al. demonstrate a van der Waals integration method to transfer large-area, high-k dielectrics on 2D semiconductors. They achieve this by cautiously selecting the sacrificial layer, i.e. PVA, which act as both a buffer layer for ALD deposition of Al₂O₃ and HfO₂, and a protection layer during transferring. Based on such integration, high-performance MoS₂ field-effect transistors can be realized, which exhibit negligible residual doping, a small subthreshold swing, and operation under low operation voltages. Furthermore, a variety of logic gates are realized by exploiting these MoS₂ field-effect transistors. The results are of high quality and should lead to a significant advancement in the 2D-dielectric integration technology. Thus, I recommend its publication after the following questions being addressed.

Response: We thank reviewer for carefully reading our manuscript and the recognition that “The results are of high quality and should lead to a significant advancement in the 2D-dielectric integration technology” and supports for its publication. We particularly appreciate the highly insightful and constructive comments from the reviewer and welcome the opportunity to address these questions and describe the revisions we have made accordingly.

Specific Comment 1. The dielectric constant of transferred HfO₂ is significantly lower than the theoretical value. What could be the reason? Is it due to the transfer process or ALD growth conditions?

Response: We thank the reviewer for this important question about the dielectric constant of HfO₂. First of all, we note the dielectric constant (κ) of our HfO₂ decreases with its thickness, dropping to a low κ value ~ 10 (with sub-10 nm thickness) and is much smaller than its theoretical value (~ 20). This behavior is consistent with previous literatures, where relatively low dielectric constant ($\kappa=7-10$) is also measured with similar sub-10 nm HfO₂ film [*Appl. Phys. Lett.* 101, 172910 (2012); *J. Electrochem. Soc.* 151, F189 (2004); *Thin Solid Films* 491, 328 (2005).], as shown in Fig. R13a below. This reduced- κ value and thickness-dependence- κ could be explained by the dielectric dead layers [*Nature* 443, 679 (2006); *Appl. Phys. Lett.* 101, 172910 (2012); *Nature* 605, 262 (2022)], where two interfacial capacitances (C_{i1} and C_{i2}) exist at the electrode/dielectric boundaries, as schematical illustrated in Fig. R13b. These interfacial capacitances typically exhibit lower κ -values (hence termed dead layers) and act as series-capacitance with the pristine bulk capacitance (C_{bulk}). Hence, the overall dielectric constant is smaller than its bulk value, and such effect is more and more pronounced with decreasing film thickness. To confirm this theory, we have conducted additional control experiment by vdW laminating thicker HfO₂ (25 nm thick). As shown in Fig. R13c, the measured dielectric constant is ~ 15 , much closer to its bulk value (~ 20). Furthermore, following the reviewer suggestions, we have also measured the properties of as-grown HfO₂ without transfer processes. As shown in Fig. R13c, the directly grown HfO₂ also decreases with reducing thickness. In particular, the dielectric constant of as-grown film is identical with film after vdW transfer lamination, further indicating our lamination process won't impact the intrinsic properties of the dielectric film.

We thank the reviewer for these questions, and we have further explained the low- κ value of HfO₂ in the revised manuscript.

[redacted]

Fig. R13. a, The EOT and dielectric constant value of HfO₂ versus its thickness. This figure is replotted from [*J. Electrochem. Soc.* 151, F189 (2004)]. **b**, Schematic of dielectric dead layer effect, which is replotted from [*Nature* 605, 262-267 (2022)]. **c**, Dielectric constant of HfO₂ with different thickness, for both direct grown HfO₂ (red line), as well as for vdW laminated HfO₂ film (blue line).

Specific Comment 2. The thinner Al₂O₃ also display very small leakage current, but the authors chose 10-nm-thick Al₂O₃ for the fabrication of MoS₂ transistors and logic gates. The rationale behind this choice should be provided.

Response: We thank the reviewer for this question. In our top-gate MoS₂ transistor, 10 nm thick Al₂O₃ dielectric is used to demonstrate the minimized doping effect during vdW lamination process. We also agree with the reviewer that “thinner Al₂O₃ also display very small leakage current” and could be used to fabricate MoS₂ transistors. To demonstrate this, we have measured top-gate transistor by laminating 4.5 nm thick Al₂O₃ as the top-dielectric. As shown in Fig. R14 below, the I_{ds} - V_{gs} transfer curve demonstrate high on-off ratio over 10⁶ and low gate leakage current of 100 fA. More importantly, the device could be operated under lower gate voltage of 0.5 V (due smaller EOT) with SS of ~70 mV/dec, suggesting the stronger gate control using thinner dielectric.

We thank the reviewer for this question, and we have included top gate MoS₂ transistor with thinner dielectrics in the revised manuscript.

Fig. R14. I_{ds} - V_{tg} transfer curves and gate leakage current of top-gate MoS₂ transistors fabricated by laminating 4.5 nm Al₂O₃, exhibiting high on/off ratio and low gate leakage current.

Specific Comment 3. Besides PVA, PMMA, and PPC, has the authors considered other sacrificial polymer layer? Is there a general selection rule?

Response: We thank the reviewer for the important question. Besides PVA, PMMA and PPC, we have also tried other commonly used polymers, including PC (polycarbonate) and PI (polyimide). Among all these polymers, PVA buffer layer is used as our buffer layer due to two general rules. First, PVA demonstrate low adhesion force towards our functionalized substrate and can be mechanically peeled-off. For other polymer buffers, they indeed show larger adhesion force and can not be easily peeled-off in wafer scale. Second, the spin-coated PVA buffer is thin enough (~9 nm) and still demonstrates atomic flat surface (0.36 nm surface roughness). For other conventional used polymer, sub-10 nm thick continuous film is hard to realize. The ultra-thin PVA thickness not only ensure successful etching using gentle plasma treatment, but also is essential to minimize the strain during the transfer process. Based on the above discussion, we

believe the low adhesion force and the small thickness are the general rules to select polymer buffers.

Specific Comment 4. Have the authors tried to apply this integration method to other 2D materials? If so, is it reproducible?

Response: Thanks for the excellent point. Within our method, the wafer-scale dielectric is mechanically laminated on top of the 2D surface and is not limited by the 2D channels. Hence, it is a universal approach and could be extended to other 2D semiconductors. To demonstrate this, we have fabricated monolayer WSe₂ transistors using Au as the contact metal. As shown in Fig. R15a, the as-fabricated WSe₂ transistor demonstrates p-type I_{ds} - V_{gs} transfer curve using 300 nm thick SiO₂ as back-gate dielectric. After vdW laminating 10 nm Al₂O₃ on top of the device, identical device behavior is observed, suggesting the dielectric integration won't impact the intrinsic behavior of WSe₂ transistors. Furthermore, we have measured the top-gate WSe₂ device properties using vdW Al₂O₃ as the top-dielectric, and the device exhibits decent high on-off ratio over 10⁶ and small hysteresis (Fig. R15b), indicating our method is a universal approach and the optimized vdW interfaces could be reproduced to other 2D channels.

We thank the reviewer for this question, and we have included the performance of top-gate WSe₂ transistors in the revised manuscript.

Fig. R15. a, I_{ds} - V_g transfer curves of WSe₂ transistor before and after integrating Al₂O₃ insulator films. **b,** I_{ds} - V_{tg} transfer curves of top-gate WSe₂ transistor fabricated by using 10 nm Al₂O₃ vdW dielectric.

Specific Comment 5. The connection and working mechanism of the NMOS logic inverter can be described in more detail.

Response: We thank the reviewer for this question. The circuit connection and the optical image of the NMOS inverter is shown in Fig. R16 below. In general, two n-type transistors (MoS₂ in our case) are connected in series, where the gate electrode and source electrode of top transistor (T1) is connected as a pull-up resistor. When a negative V_{in} is applied, the bottom transistor (T2) is switched-off and the device generate a high V_{out} . On the other hand, when a positive V_{in} is applied, the bottom transistor is switched-on and the V_{out} is low, leading to the function of inverter, as shown in Fig. R16. This NMOS inverter work mechanism is consistent with previous literatures [*Nature*, 562, 254 (2018); *IEEE EDL*, 15, 455 (1994)].

We thank the reviewer for this question, and we have further explained the connection and working mechanism of NMOS inverter in the revised manuscript.

Fig. R16. a, Schematics of NOMS inverter logic. T1: transistor 1, T2: transistor 2. **b**, Optical image of NMOS logic inverter constructed by connecting two MoS₂ transistors in series.

Specific Comment 6. The authors have measured quite a few devices. So error bars could be added to Fig. 2c,2e and Fig. 3f to make them more sound.

Response: Thank and we have now included the error bars in the revised Fig. 2c, Fig. 2e and Fig. 3f.

References

1. Illarionov, Y. Y. et al. Ultrathin calcium fluoride insulators for two-dimensional field-effect transistors. *Nat. Electron.* **2**, 230-235 (2019).
2. Huang, J.-K. et al. High- κ perovskite membranes as insulators for two-dimensional transistors. *Nature* **605**, 262–267 (2022).
3. Zhang, Y. et al. A single-crystalline native dielectric for two-dimensional semiconductors with an equivalent oxide thickness below 0.5 nm. *Nat. Electron.* **5**, 643-649 (2022).
4. Li, W. et al. Uniform and ultrathin high- κ gate dielectrics for two-dimensional electronic devices. *Nat. Electron.* **2**, 563–571 (2019).
5. Li, T. et al. Electrical performance of multilayer MoS₂ transistors on high- κ Al₂O₃ coated Si substrates. *AIP Adv.* **5**, 057102 (2015).
6. Wen, M. et al. Effects of annealing on electrical performance of multilayer MoS₂ transistors with atomic layer deposited HfO₂ gate dielectric. *Appl. Phys. Express* **9**, 095202 (2016).
7. Liu, K. et al. A wafer-scale van der Waals dielectric made from an inorganic molecular crystal film. *Nat. Electron.* **4**, 906-913 (2021).
8. Chang, W. H. et al. ALD-ZrO₂ gate dielectric with suppressed interfacial oxidation for high performance MoS₂ top gate MOSFETs. *Jpn. J. Appl. Phys.* **60**, SBBH03 (2021).

REVIEWERS' COMMENTS

Reviewer #1 (Remarks to the Author):

I would like to thank the authors for their diligent response to my concerns. From my perspective, the papers can be basically accepted as is.

Reviewer #2 (Remarks to the Author):

Thanks to the author for his efforts, my concerns have been adequately addressed, and I agree that the current manuscript will be published in Nature communication

Reviewer #3 (Remarks to the Author):

My questions have been adequately addressed with the support of additional experimental results. Corresponding modifications have been made to the manuscript. Therefore, I recommend its publication in the current form.